# Renin Feedback Is an Independent Predictor of Outcome in HFpEF

**DOI:** 10.3390/jpm11050370

**Published:** 2021-05-03

**Authors:** Christina Binder, Marko Poglitsch, Franz Duca, René Rettl, Theresa Marie Dachs, Daniel Dalos, Lore Schrutka, Benjamin Seirer, Luciana Camuz Ligios, Christophe Capelle, Roza Badr Eslam, Hong Qin, Christian Hengstenberg, Diana Bonderman

**Affiliations:** 1Division of Cardiology, Medical University of Vienna, 1090 Wien, Austria; christina.binder@meduniwien.ac.at (C.B.); franz.duca@meduniwien.ac.at (F.D.); rene.rettl@meduniwien.ac.at (R.R.); theresa-marie.dachs@meduniwien.ac.at (T.M.D.); daniel.dalos@meduniwien.ac.at (D.D.); lore.schrutka@meduniwien.ac.at (L.S.); benjamin.seirer@meduniwien.ac.at (B.S.); luciana.camuzligios@meduniwien.ac.at (L.C.L.); christophe.capelle@meduniwien.ac.at (C.C.); roza.badr-eslam@meduniwien.ac.at (R.B.E.); hong.qin@meduniwien.ac.at (H.Q.); christian.hengstenberg@meduniwien.ac.at (C.H.); 2Attoquant Diagnostics, 1030 Vienna, Austria; marko.poglitsch@attoquant.com

**Keywords:** heart failure, renin, angiotensin, RAAS, outcome

## Abstract

Drugs which interact with the renin angiotensin aldosterone system (RAAS) aim to reduce the negative effects of angiotensin (Ang) II. Treatment with these drugs anticipate a compensatory up-regulation of renin; however, it has been shown that there is a large variability in circulating plasma renin (PRA), even in patients with optimal medical therapy in patients with heart failure (HF) with reduced ejection fraction (HFrEF). Our aim was to measure plasma renin activity (PRA-S), its response to RAAS inhibitor (RAASi) therapies and its effects on outcome in patients with HF with preserved ejection fraction (HFpEF). For this purpose, 150 HFpEF patients were included into a prospective single-center registry. Equilibrium (eq) angiotensin metabolites were measured from serum samples using mass spectroscopy. PRA-S (eqAng I + eqAng II) was calculated and compared in respect to the primary endpoint defined as all-cause death. PRA-S in patients with RAASi therapy was not significantly higher than in patients without RAASi (*p* = 0.262). Even after adjusting for confounding factors, PRA-S remained predictive for all-cause death in the multivariable model with a hazard ratio of 2.14 (95%CI 1.20–3.82, *p* = 0.010). We conclude that high PRA-S is associated with poor prognosis in patients with HFpEF, regardless of RAASi treatment, which could ultimately result in hyperactivated RAAS and consecutive negative effects on the cardiovascular and renal system, leading to poor outcome in patients with HFpEF.

## 1. Introduction

### 1.1. The Renin Angiotensin Aldosterone System and Its Effects

The renin angiotensin aldosterone system (RAAS) is a well-studied cascade, which acts as a regulator of blood pressure and fluid homeostasis. In the RAAS pathway, angiotensin (Ang) I is generated by enzymatic cleavage from hepatic angiotensinogen via renin secreted by the juxtaglomerular cells of the kidney. It is then converted to Ang II by the angiotensin converting enzyme (ACE) [1]. An overshoot of RAAS activity at the level of renin induces multiple molecular pathways, which exert negative effects on the cardiovascular and renal system. These include direct vasoconstriction via Ang II, fluid retention via Ang II-induced aldosterone release and many others [2]. The alternative RAAS is an additional pathway, which counteracts the undesirable effects of Ang II, mostly via Ang 1–7 and is closely intertwined with the classical RAAS system and its mediators. Its significance is not yet fully understood and is a source for future investigations.

### 1.2. The Renin Angiotensin Aldosterone System in Patients with Heart Failure

For many years, the significance of the RAAS has been appreciated in the pathophysiology of heart failure (HF). Drugs which interact with the RAAS have become an important pillar of HF therapies in patients with reduced ejection fraction (HFrEF) [3,4,5]. High levels of plasma renin translate into higher levels of Ang II with all its well-known detrimental effects on the cardiovascular system including vasoconstriction, endothelial dysfunction, induction of fibrosis and many others [6,7].

In principle, the administration of RAAS inhibitors (RAASi) such as ACE-inhibitors (ACEi) or angiotensin-receptor-blockers (ARBs) leads to a significant change in plasma angiotensin profiles, driven by direct pharmacodynamic drug effects. However, it is essential to understand that these changes in return cause a compensatory up-regulation of renin via a renal feedback mechanism involving renal angiotensin II type 1 receptor (AT1R) signaling [8]. An effective RAAS blockade is crucial to improve outcomes by preventing the progression of cardiovascular disease. However, this may be counteracted by patient-specific features of compensatory mechanisms.

### 1.3. Plasma Renin as a Predictor of Outcome in Heart Failure

Plasma RAAS components have repeatedly been described as predictors of outcome in HF patients [9]. Although there is an abundance of evidence supporting the importance of the RAAS in HFrEF, respective data in patients with HF and preserved ejection fraction (HFpEF) are much more scarce. The conversion of angiotensinogen to Ang I via renin represents the rate-limiting step of the RAAS cascade and can therefore be seen as a marker of RAAS activation. Plasma renin activity (PRA) can accurately be expressed as the sum of equilibrium levels of Ang I and Ang II (PRA-S) [10,11] even without directly measuring plasma renin concentration. PRA correlates with blood pressure and pre-treatment plasma renin levels were able to predict the response to blood pressure medication [12,13]. Present data also suggest that patients with higher levels of plasma renin have an increased risk for ischemic events and congestive HF, even irrespective of blood pressure [14]. Pavo et al. have recently described low- and high-renin HFrEF phenotypes, indicating that patients react to RAAS inhibitors with a different extent of renin up-regulation [15]. This might ultimately lead to an Ang II overshoot that cannot be blocked sufficiently by standard doses of ACEi or ARBs, resulting in suboptimal HF treatment. Thus far, these observations have only been described in patients with HFrEF. It is not known if patients with HFpEF show similar responses to RAAS inhibitors. We conducted this study with the intention to describe PRA-S and its response to RAAS inhibitors, as well as its effects on outcome in patients with HFpEF.

## 2. Materials and Methods

### 2.1. Study Design

This study prospectively examined the effects of RAAS-associated biomarkers on outcome in patients with HFpEF. Consecutive patients were included into a prospective clinical registry at the Division of Cardiology of the Medical University of Vienna, a national HFpEF referral center at our high-volume HF outpatient clinic. All patients gave written informed consent before study inclusion. The protocol is in line with the Declaration of Helsinki and was approved by the local Ethics Committee (EK #796/2010). Baseline assessment included clinical examination, the documentation of patient history and drug therapy. Imaging and laboratory testing were performed, including the measurement of serum equilibrium levels of angiotensins and aldosterone. Following baseline tests, mineralocorticoid antagonists (MRAs) were initiated in all patients during follow-up visits, if potassium and creatinine levels allowed.

### 2.2. Transthoracic Echocardiography

Transthoracic echocardiography (TTE) was used to confirm the diagnosis of HFpEF and was performed by board-certified physicians on high-end machines (GE Vivid 7 and Vivid E9; GE Healthcare, Wauwatosa, WI, USA) in accordance with current guidelines [16,17]. TTE images were recorded within a 2-month timeframe from the assessment of clinical parameters and biomarker sampling and analyzed using a modern offline clinical workstation equipped with dedicated software (EchoPAC; GE Healthcare, Wauwatosa, WI, USA).

### 2.3. Clinical Definitions

The diagnosis of HFpEF was made according to current consensus statements of the European Society of Cardiology [18] and the guidelines proposed by the American Heart Association [19]. Therefore, patients were included when the following criteria were present: (1) signs and symptoms of HF, (2) left ventricular ejection fraction ≥ 50%, (3) N-terminal-pro brain natriuretic peptide (NT-pro BNP) levels exceeding 220 pg/mL and (4) evidence of left ventricular (LV) diastolic dysfunction and/or structural heart disease with either left atrial enlargement (LA volume index > 34 mL/m^2^) or LV hypertrophy (interventricular septum thickness ≥ 11 mm LV mass index ≥ 115 g/m^2^ in males and ≥95 g/m^2^ in females). Patients were not included, when they had significant coronary artery disease, confirmed by left heart catheterization and defined as a visual stenosis of more than 50% in one of the main vessels and/or over 70% in a distal vessel.

### 2.4. Clinical Outcomes

Clinical outcomes were ascertained by follow-up at our dedicated HFpEF outpatient clinic as well as clinical records or phone calls in the case of patient immobility. The primary outcome parameter was defined as death by any cause. When an event occurred, local and external medical records were carefully screened and the cause of death was reviewed by a clinical adjudication committee of board—certified cardiology specialists (D.B., D.D, and R.B.).

### 2.5. Blood Sampling

Blood sampling was performed at baseline and analyzed according to local laboratory standards. In addition to routinely performed laboratory parameters, serum samples were drawn at study baseline. Samples were centrifuged at 1300 rpm for 10 min and frozen at −80°C for biomarker assessment.

### 2.6. Renin Angiotensin Aldosterone System (RAAS) Triple Analysis

Equilibrium concentrations of Ang I (eqAng I), Ang II (eqAng II) and aldosterone levels were measured following ex vivo equilibration and subsequent stabilization of conditioned serum samples as described previously [20,21]. Stabilized samples were spiked with stable isotope-labelled internal standards for individual angiotensins and aldosterone at concentrations of 200 pg/mL and 500 pg/mL, respectively. Following C18-based solid-phase-extraction and fractionated elution of analytes, samples were subjected to liquid chromatography tandem mass spectrometry (LC-MS/MS) analysis using a reversed-phase analytical column (Acquity UPLC^®^, C18, Waters, Milford, MA, USA) operating in line with a XEVO TQ-S triple quadruple mass spectrometer (Waters Xevo TQ/S, Milford, MA, USA) in multiple reaction monitoring mode. Internal standards were used to correct for analyte recovery across the sample preparation procedure in each individual sample. Analyte concentrations were calculated from integrated chromatograms considering the corresponding response factors determined in appropriate calibration curves in serum matrix, on the condition that integrated signals exceeded a signal-to-noise ratio of 10. The lower level of quantification (LLOQ) for serum equilibrium Ang I, Ang II and aldosterone were 5 pmol/L, 2 pmol/L and 14 pmol/L, respectively. Based on measured values for eqAng I, eqAng II and aldosterone, angiotensin-based markers for renin activity (PRA-S), ACE activity (ACE-S) and adrenal AT1-receptor function (AA2-Ratio) were calculated as shown in Table 1. Figure 1 shows RAAS Triple-A analysis and angiotensin metabolism.

### 2.7. Statistical Analysis

Categorical variables are shown as numbers and percentages, and continuous variables are expressed in median and interquartile range. A *p*-value < 0.05 was considered statistically significant. IBM SPSS version 24 (SPSS Inc., Chicago, IL, USA) was used for all statistical analyses.

Patients were divided by median PRA-S, calculated as (eqAng I+ eqAng II) for baseline assessment. Baseline characteristics of the total study population, as well as for the aforementioned groups were computed using one-way ANOVA or the Chi-square test as appropriate. Angiotensin profiles were displayed for patient groups receiving different RAAS-inhibiting therapies and compared using Mann–Whitney U-test. To depict angiotensin concentrations and compare them between different groups of RAAS-inhibiting therapy, graphs were created using the median of each angiotensin concentration.

For Kaplan–Meier analysis, “high versus low” PRA groups were categorized according to median PRA-S. To assess the effects of PRA-S on outcomes, Kaplan–Meier plots were calculated for the endpoint all-cause death. Univariable Cox regression analysis was performed for all parameters. To account for possible confounding effects, a multivariable stepwise-forward model was applied by entering predictive parameters separately for clinical, laboratory and imaging parameters from the univariable model at a significance level of 0.05.

## 3. Results

### 3.1. Study Population

Between December 2010 and December 2018, a total of 159 patients were considered eligible for this study. Four patients were excluded due to hemolytic serum samples, which could not be analyzed. Further, five patients were excluded due to diagnosis of cardiac amyloidosis (*n* = 2), reduced left ventricular ejection fraction (*n* = 2) and severe aortic stenosis (*n* = 1) during follow-up, resulting in a final patient population of 150 patients (Appendix A). Of these, 45 patients (30.0%) received ACEi, 66 patients (23.3%) received ARB and 53 patients (35.3%) received MRA at baseline. A total of 35 patients (24.0%) were on a combination of either ACEi and MRA (*n* = 17) or ARB and MRA (*n* = 18). An additional 18 patients (12.0 %) were only on MRA at the time of assessment. Figure 2, Figure 3 and Figure 4 visually depict the diversity of angiotensin profiles in patients without RAAS-inhibiting therapy, patients receiving ACEi and patients on ARBs. Table 2 shows detailed angiotensin profiles according to therapy groups (ACEi, ARB and no RAASi).

Drug effects were clearly reflected by serum levels of eqAng I and eqAng II (Figure 2). In most patients on ACEi, ACE-S was significantly suppressed with higher levels of eqAng I and suppressed eqAng II. However, there was a subgroup of patients on ACEi without ACE-S suppression. Patients on ARBs had high levels of eqAng I and eqAng II, while ACE-S remained unaffected (Table 2, Figure 2A and Appendix A). Furthermore, the AA2-Ratio was lower in ARB-treated patients, indicating the suppression of adrenal aldosterone secretion by efficient blockage of adrenal Ang II receptor type 1 (AT1R) blockade. However, there was no difference in PRA-S between patients on ACEi or ARBs (*p* = 0.714). Patients treated with MRA at baseline had higher levels of aldosterone, Ang I and Ang II, and therefore also higher levels of PRA-S. Appendix A show detailed comparisons between all therapy groups separately.

### 3.2. Baseline Characteristics According to Plasma Renin Activity (PRA-S)

Baseline characteristics of the total study population, as well as for PRA-S groups are shown in Table 3. The median age among the entire study population was 72.0 years (IQR 67.0–76.0) and there were no significant differences in ACEi or ARB treatment between the patient groups. A vast majority of patients had arterial hypertension (*n* = 140, 93.3%) and more than half of the total study population presented with atrial fibrillation (*n* = 84, 56.0%). Patients with higher PRA-S presented with lower blood pressure and more pronounced renal impairment. Median NT-pro BNP levels were markedly elevated among the total study population (1050 pg/mL, IQR 507–1788) and there was no difference between the PRA-S groups.

Patients with higher PRA-S had higher serum eqAng I and eqAng II and lower ACE-S, as well as eqAng 1–5. However, among patients without RAAS inhibitors, we observed patients with elevated PRA-S, while others showed low PRA-S (Figure 3), indicating a low and high renin phenotype in RAAS inhibitor naïve patients.

During a mean follow-up time of 40.9 ± 27.7 months, 30 (20.0%) patients died and 48 (32.0%) were hospitalized for HF. Of all deaths, 13 (43.3%) were classified as cardiac. Eleven deaths were attributed to non-cardiac causes. These included end-stage cancer (*n* = 4), sepsis (*n* = 2), pulmonary embolism (*n* = 2), iatrogenic causes (*n* = 2) and liver failure (*n* = 1). An additional six deaths were verified; however, the cause of death could not be ascertained, due to missing documentation and the inability to contact relatives of the deceased individuals. There was no difference in outcomes regarding therapy with ACEi, ARBs or no RAASi (Figure 5A). High levels of eqAng II, which has a multitude of detrimental effects on the cardiovascular system, predicted adverse outcome (Table 4, Figure 5B). However, Cox regression, as well as Kaplan–Meier analysis also demonstrated that higher PRA-S was predictive for poor outcome in HFpEF patients. This was true regardless of RAASi therapy. After adjusting for other outcome-associated laboratory values, which were statistically significant with a *p*-value < 0.05 in the univariable model, including NT-pro-BNP, eqAng I, eqAng II as well as presence of and type of RAASi, only PRA-S remained independently predictive for outcome within the category of laboratory parameters with an adjusted hazard ratio of 2.142 (95%CI 1.203–3.815, *p* = 0.010). Our data show that higher levels of eqAng II, as well as higher PRA-S are associated with poor outcome (Figure 5B,C).

## 4. Discussion

The present study shows angiotensin profiles in patients with HFpEF under different RAASi therapies. We identified PRA-S as a marker of RAAS activity as well as a predictor for all-cause death.

While there are extensive data on the positive effects of RAAS-inhibiting effects on outcome in patients with HFrEF, it is not yet understood why blocking the RAAS is not effective in preventing adverse outcomes in HFpEF. It has been discussed that flaws in the study designs of HFpEF trials may have influenced the results and it has been postulated that HFpEF patients at the low ejection fraction spectrum may benefit from these RAASi. However, the question remains, why the effectiveness of RAASi is lacking in HFpEF.

Since arterial hypertension represents a crucial risk factor for the development of HFpEF, it seems counterintuitive that RAASi would not be effective in HFpEF [22]. Even though RAASi do not seem to improve outcome in patients with evident HFpEF, they may play a role in preventing HFpEF. Therefore, gaining insights to individual RAAS profiles in these patients may help in finding treatment regimens for arterial hypertension in order to obtain optimal blood pressure control [23]. PRA-S has been validated previously by Burello et al. as a reliable surrogate for plasma renin activity in patients with and without RAASi treatment. As shown in Figure 1 and Table 2, treatment with ACEi anticipates a significant elevation of Ang I, while patients treated with ARBs show higher levels of both Ang I and Ang II. Consequently, using the sum of these two parameters reflects the overall effect on renin up-regulation, independent of the type of RAAS blockade. Therefore, PRA-S has been proposed as a promising parameter for diagnostic screening of primary aldosteronism, as well as for therapy response [15,24].

In our patient cohort, we saw that patients with higher PRA-S also presented with lower blood pressure. This indicates that RAASi therapies seemed to have an effect; however, this did not translate into better outcome (Figure 5A). Instead, these findings might point to different underlying molecular mechanisms of blood pressure regulation and target organ damage, potentially involving local tissue RAAS components and drug access to these compartments. We also observed that higher levels of PRA-S were associated with more impaired kidney function. This is not surprising, since a highly active RAAS system leads to the impairment of renal function via a multitude of Ang-mediated pathways, including vasoconstriction and reduced renal blood flow, induction of inflammatory processes [25] and increased fibrotic response [26].

Interestingly, in patients who were treatment naïve to RAASi, a subgroup showed high levels of PRA-S. This was an effect which was not influenced by concomitant beta-blocker intake (Table 3). As previously described in patients with HFrEF by Pavo et al. [15,21], a “low and high renin phenotype” seems to also exist in patients with HFpEF. Similar findings were noted by Vergaro et al., who presented a biohumoral panel, which included PRA, aldosterone and norepinephrine. The authors showed that an elevation of these parameters correlated with adverse outcomes. These effects were not only found in patients with HFrEF, but across the entire heart failure spectrum irrespective of ejection fraction [27]. Their results are in line with our data regarding outcome in patients with high PRA-S. Our study provides further data, not only on PRA and aldosterone, but on entire angiotensin profiles in separate subsets of patients characterized by type of RAASi therapy. The inter-individual variability of RAAS biomarkers within treatment groups was large, reaching up to three orders of magnitude. Of note, renin secretion, being the major driver of angiotensin formation, is subject to modulation by multiple factors, including RAASi therapy, renal perfusion, salt intake and also genetic factors [28,29,30,31]. Considering our results, it seems essential to study factors which influence PRA-S, as well as identify interventions, which may lower PRA-S to potentially improve outcomes.

We suspect that negative Ang-effects (i.e., induction of fibrosis via aldosterone, fluid overload) may not be inhibited efficiently in some patients receiving ACEi or ARB. This could be caused by a compensatory up-regulation of the RAAS system, ultimately leading to higher levels of Ang II than would be necessary to prevent adverse outcomes. It is likely that complex RAAS-mediated interactions of the cardio–renal system, which may also be triggered by inadequate dosage of RAASi, could lead to an overshoot of renin activity. This could ultimately lead to an increase in pro-fibrotic signals and could further promote myocardial stiffness and impaired relaxation. This could be especially detrimental in HFpEF, as myocardial fibrosis is a central mechanism in the pathogenesis of this disease. Assessment of individual PRA-S, ACE-S and AA2-Ratio values on treatment by RAAS Triple-A analysis may be key to finding the optimal dose in this patient population, assuring sustained pharmacodynamic efficacy and adequate individualized dosage of therapeutic RAAS blockade.

### Limitations

Several limitations exist regarding the characteristics of the study population and methodology. We are aware of the limitations that go along with the single-center design of this study. On the other hand, advantages of this study lie in the consistency of diagnostic workup and treatment and follow-up. Due to the relatively small size of our study cohort, the number of clinical events is limited. The limited number of events in the present study does not allow extensive Cox regression analysis. Our multivariable Cox regression model included five variables compared to 30 outcome events. Therefore, these results must be interpreted with caution, as low event per variable ratios may affect type I errors and may cause bias away from the null hypothesis [32]. Furthermore, while all-cause death is an important endpoint, it should be noted that quality of life is also an important outcome parameter, which was not assessed in this study. The improvement of quality of life (i.e., by controlling hypertensive episodes or providing adequate diuresis) could also potentially be an incentive to initiate RAASi treatment. While concomitant RAASi therapies were assessed at the time of blood sampling, we did not consider drug dosage, treatment compliance and the duration of RAASi treatment. RAAS profiles suggest that compliance was presumably an issue, as shown by unaffected ACE-S in a subset of patients who were allegedly on ACEi (Figure 3). Even if so far, large trials have failed to give definite answers on the effects of MRA on outcome in HFpEF [33], the additional administration of MRA during the follow-up time may have had an impact on the presented outcomes. Furthermore, MRA treatment could have impacted RAAS profiles in the sense that incomplete ACEi or ARB effects might not be aldosterone-mediated, but rather caused by direct vasoconstriction. Since angiotensin measurements were only performed once at baseline, this study does not give insights into the dynamics of angiotensins over a time span of disease progression. Concomitant conditions, such as diabetes and renal failure, potentially could cause complex interactions, which might have influenced the results by enhancing ACE2 activity. In our opinion, these conditions should be seen as a part of the HFpEF phenotype rather than individual diseases.

## 5. Conclusions

The present study shows that high levels of eqAng II and PRA-S, even despite RAASi treatment, are associated with poor prognosis in patients with HFpEF. This may ultimately result in negative effects on the cardiovascular and renal system.

## Figures and Tables

**Figure 1 jpm-11-00370-f001:**
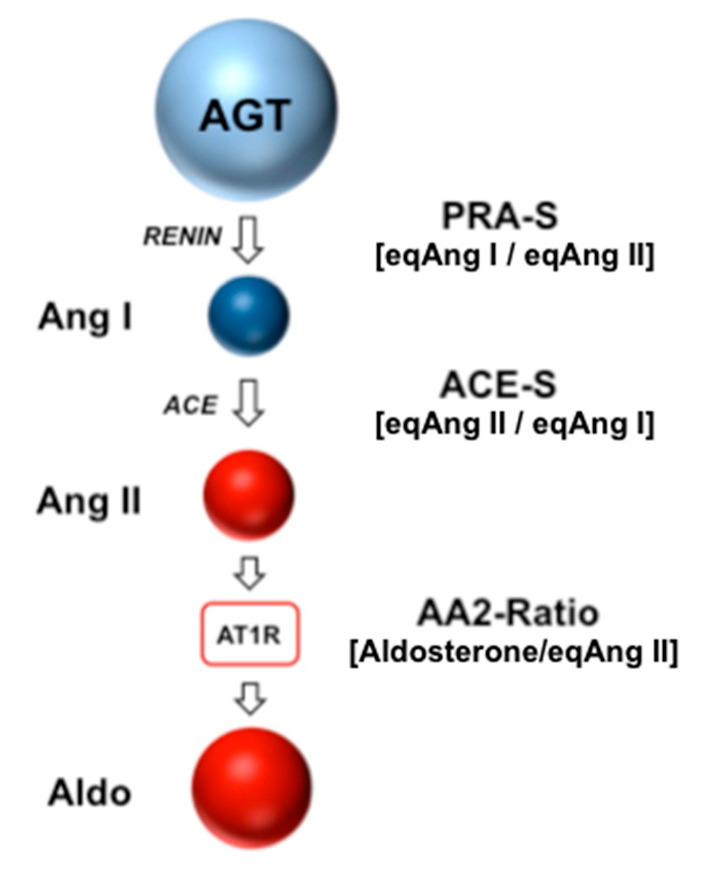
Angiotensin metabolism and RAAS Triple-A analysis. Angiotensinogen is produced in the liver and converted to angiotensin I by enzymatic cleavage via renin. Angiotensin I is further metabolized to angiotensin II by angiotensin converting enzyme, which binds to the angiotensin II type 1 receptor (AT1R), inducing vasoconstriction and secretion of aldosterone from the adrenal cortex. In RAAS Triple-A analysis, angiotensin-based markers for plasma renin activity (PRA-S), ACE activity (ACE-S) and adrenal AT1R signaling (AA2-Ratio) are calculated as indicated.

**Figure 2 jpm-11-00370-f002:**
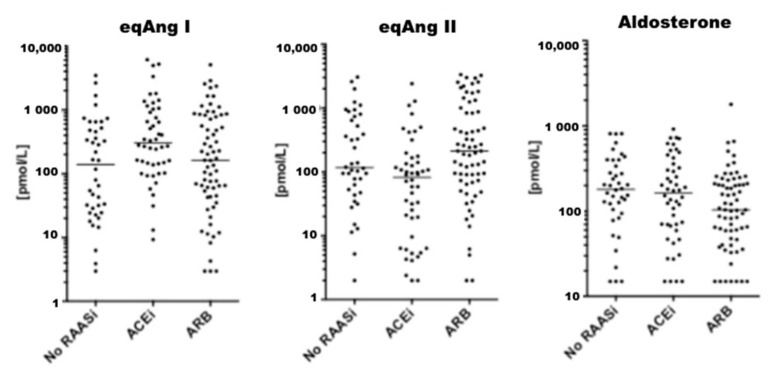
Value distribution for Ang I, Ang II and Aldosterone. Graphs show individual values for indicated analytes. The median is indicated as a horizontal line.

**Figure 3 jpm-11-00370-f003:**
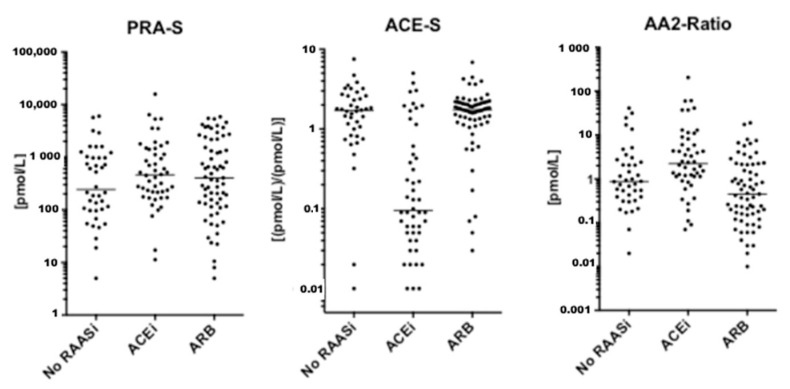
Value distribution for PRA-S, ACE-S and AA2-Ratio. Graphs show individual values for indicated analytes. The median is indicated as a horizontal line.

**Figure 4 jpm-11-00370-f004:**
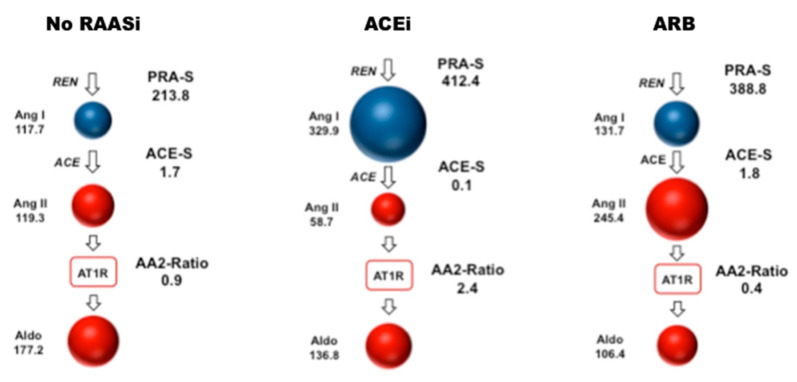
RAAS Triple-A illustrations visualizing serum concentrations of Ang I, Ang II and aldosterone. Numbers refer to median serum equilibrium concentrations in (pmol/L). Median valued for patient groups without renin angiotensin-inhibiting therapy (*n* = 39), on ACEi therapy (*n* = 45) and on ARB treatment (*n* = 66) are shown as indicated. Corresponding median values for PRA-S, ACE-S and AA2-Ratio are shown in (pmol/L), ((pmol/L)/(pmol/L)) and ((pmol/L)/(pmol/L)), respectively. RAASi indicates renin angiotensin system inhibitors; ACEi, angiotensin converting enzyme inhibitors; ARB, angiotensin receptor blockers; Ang, angiotensin; REN, renin; ACE, angiotensin converting enzyme; AT1R, angiotensin II type 1 receptor; Aldo, aldosterone; PRA-S: plasma renin activity in (pmol/L); ACE-S: ACE activity in ((pmol/L)/(pmol/L)); AA2-Ratio: Aldosterone-to-Ang II ratio in ((pmol/L)/(pmol/L)).

**Figure 5 jpm-11-00370-f005:**
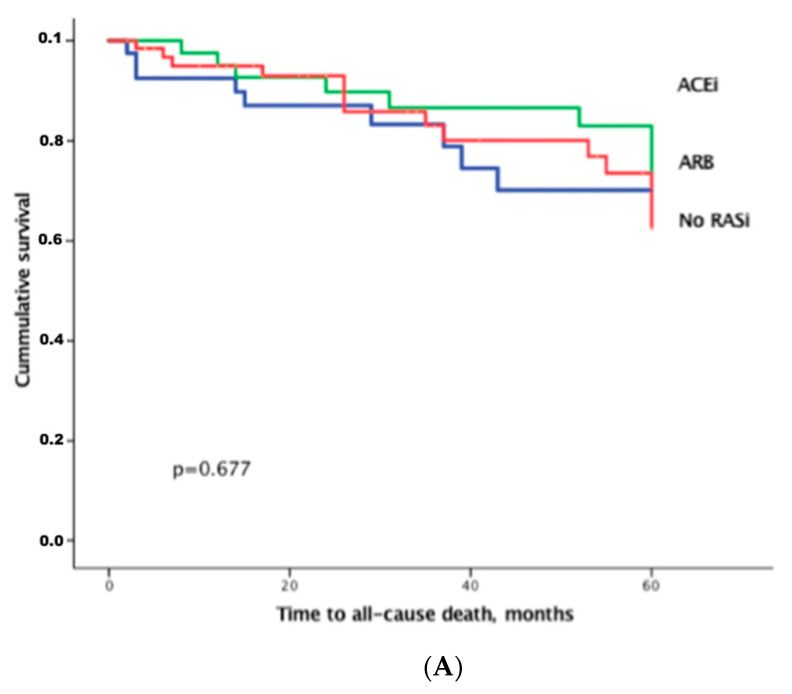
(**A**) Kaplan–Meier analysis for all-cause death between therapy groups in the total study cohort (*n* = 150). (**B**) Kaplan–Meier analysis for time to all-cause death, shown for patients with high- versus low levels of Angiotensin II, separated by median Angiotensin II levels. (**C**) Kaplan–Meier analysis for time to all-cause death, shown for patients with high versus low plasma renin activity (PRA), separated by median plasma renin levels. ACEi indicates angiotensin converting enzyme inhibitors; ARB, angiotensin receptor blockers; RASi, renin angiotensin system inhibitors, which includes ACEi and ARB.

**Table 1 jpm-11-00370-t001:** Calculated parameters of the renin angiotensin aldosterone system (Triple A Test^TM^).

AA2-Ratio.	(Aldo)/(Ang II)	Adrenal AT1R signaling
ACE-S	(Ang II)/(Ang I)	Angiotensin-based ACE activity
PRA-S	(Ang I + Ang II)	Angiotensin-based renin activity

Aldo indicates aldosterone; Ang, angiotensin; ACE, angiotensin converting enzyme.

**Table 2 jpm-11-00370-t002:** Renin angiotensin aldosterone system (RAAS) profiles in patients with angiotensin converting enzyme inhibitors (ACEi), angiotensin receptor blockers (ARB) and without renin angiotensin aldosterone system inhibitors (RAASi).

RAAS Triple-A Analysis	ACEi (*n* = 45)	ARB (*n* = 66)	No RAASi (*n* = 39)
eqAng I, pmol/L	329.9 (149.3–916.5) *,†	131.7 (45.2–654.6)	117.7 (24.5–479.1)
eqAng II, pmol/L	58.7 (9.6–117.4) *,†	245.4 (90.6–873.2)	119.3 (53.1–739.4)
Aldosterone, pmol/L	136.8 (59.4–303.9)	106.4 (59.3–221.6)	177.2 (94.3–338.6)
AA2-Ratio, (pmol/L)/(pmol/L)	2.4 (1.1–8.1) *,†	0.4 (0.2–1.4)	0.9 (0.4–2.4)
PRA-S, pmol/L	412.4 (215.0–1119.5)	388.8 (130.9–1533.7)	213.8 (96.5–1024.0)
ACE-S, (pmol/L)/(pmol/L)	0.1 (0.1–0.3) *,†	1.8 (1.4–2.2)	1.7 (1.0–2.6)

All values are given as median and interquartile ranges * significant against no RAASi at a level of <0.05. † significant against ARB at a level of <0.05. eqAng indicates equilibrium levels of angiotensin; AA2, aldosterone/angiotensin II ratio; PRA-S, plasma renin activity; ACE-S, angiotensin converting enzyme activity.

**Table 3 jpm-11-00370-t003:** Baseline characteristics of the entire study population as well as for patients classified by plasma renin activity.

	Total Study Population (*n* = 150)	PRA-S < 408.65 pmol (*n* = 75)	PRA-S ≥ 408.65 pmol (*n* = 75)	*p*-Value
***Clinical parameters***				
Age years	72.0 (67.0–76.0)	72.0 (67.5–77.0)	72.0 (67.0–76.0)	0.589
Female	108 (72.0)	53 (70.7)	55 (73.3)	0.857
BMI, kg/m^2^	29.3 (25.1–33.9)	29.0 (24.6–34.0)	29.5 (25.8–33.8)	0.995
Systolic BP, mmHg	141 (127–158)	147 (134–160)	135 (122–154)	0.011
Diastolic BP, mmHg	80 (73–90)	82 (75–90)	80 (76–87)	0.144
***Therapy***				
ACE inhibitors	45 (30.0)	21 (28.0)	24 (32.0)	0.486
ARB	66 (44.0)	34 (45.3)	32 (42.7)	0.871
MRA	53 (35.3)	16 (21.3)	37 (50.7)	<0.001
Beta blockers	112 (74.7)	57 (76.1)	55 (73.3)	0.618
***Comorbidities***				
Arterial hypertension	140 (93.3)	69 (92.0)	71 (94.7)	0.672
Diabetes mellitus	49 (32.7)	20 (26.7)	29 (38.7)	0.094
Atrial fibrillation	84 (56.0)	37 (49.3)	47 (62.7)	0.104
Coronary artery disease	32 (21.3)	15 (20.0)	17 (22.7)	0.754
***Laboratory parameters***				
NT-proBNP, pg/mL	1050 (507–1788)	939 (525–1743)	1160 (456–1959)	0.538
eGFR, ml/min/1.73 m^2^	56.0 (43.7–72.3)	60.7 (48.7–73.8)	51.6 (40.3–68.6)	0.041
GGT, U/l	33.0 (21.0–66.0)	32.0 (17.0–66.0)	36.5 (23.5–76.0)	0.234
eqAng I, pmol/L	188.5 (52.2–654.8)	54.6 (20.9–105.4)	715.7 (359.7–1288.3)	<0.001
eqAng II, pmol/L	119.6 (43.1–449.4)	58.7 (19.0–106.8)	442.0 (166.3–1236.9)	<0.001
Aldosterone, pmol/L	141.6 (61.9–254.1)	104.7 (47.7–202.4)	185.8 (70.1–399.9)	0.001
eqAng 1–7, pmol/L	3.9 (2.5–16.7)	2.5 (2.5–2.9)	16.5 (5.5–46.5)	<0.001
eqAng 1–5, pmol/L	13.2 (2.8–56.2)	3.2 (2.0–11.3)	48.2 (14.6–105.1)	<0.001
AA2-Ratio, (pmol/L)//pmol/L)	0.9 (0.3–3.0)	2.8 (0.9–6.6)	1.0 (0.4–2.1)	<0.001
ACE-S, (pmol/L)/(pmol/L)	1.5 (0.3–2.1)	1.7 (0.7–2.8)	1.1 (0.2–1.9)	0.049
***Echo parameters***				
LV diameter, mm	44.0 (40.0–48.0)	45.0 (40.0–48.0)	44.0 (40.0–47.5)	0.631
LVEF, %	58.0 (53.0–65.0)	60.0 (55.0–63.0)	56.0 (51.0–65.0)	0.166
IVS, mm	12.0 (11.0–13.0)	12.0 (11.0–14.0)	12.0 (11.0–13.0)	0.566
LV mass index, g/m^2^	97.5 (79.0–117.0)	97.0 (82.0–115.0)	98.0 (78.0–118.0)	0.834
LA length, mm	61.0 (57.0–66.0)	60.0 (56.5–65.0)	63.0 (58.0–67.0)	0.085
RV diameter, mm	37.0 (32.0–41.0)	36.0 (31.0–40.0)	38.5 (33.0–43.0)	0.055
RA length, mm	60.5 (56.0–68.0)	59.0 (56.0–64.0)	62.0 (57.0–69.0)	0.052
E/A, ratio	1.3 (0.9–1.9)	1.4 (1.0–1.9)	1.1 (0.8–1.9)	0.313
Average e’, m/s	0.08 (0.06–0.10)	0.09 (0.07–0.10)	0.07 (0.06–0.09)	0.015
E/e’, ratio	13.5 (10.7–18.0)	12.5 (11.0–17.5)	15.8 (9.1–20.0)	0.476
TAPSE, mm	18.0 (15.0–21.0)	19.0 (17.0–21.5)	17.0 (15.0–21.0)	0.128
sPAP, mmHg	58.5 (46.0–74.0)	60.0 (46.0–74.0)	57.5 (46.0–74.5)	0.658

Continuous variables are shown as median and interquartile range, categorical variables are given as numbers and percentages. PRA-S indicates plasma renin activity; BMI, body mass index, BP, blood pressure; ACE, angiotensin converting enzyme; ARB, angiotensin receptor blocker; MRA, mineralocorticoid receptor antagonist; NT-proBNP, N-terminal pro-brain natriuretic peptide; eGFR, glomerular filtration rate; GGT, gamma-glutamyltransferase; eqAng, equilibrium levels of angiotensin; AA2-Ratio, aldosterone/angiotensin II ratio; ACE-S, angiotensin converting enzyme activity; LV, left ventricle; LVEF, left ventricular ejection fraction; IVS, interventricular septum thickness; LA, left atrium; RV, right ventricle; RA, right atrium; TAPSE, tricuspid annular plane systolic excursion and sPAP, systolic pulmonary artery pressure.

**Table 4 jpm-11-00370-t004:** Univariate and multivariate Cox regression analysis for the entire study population for the primary endpoint of all-cause death.

	Univariate HR (95% CI)	*p*-Value	Multivariate HR (95% CI)	*p*-Value
***Clinical parameters***				
Age years	0.779 (0.963–1.051)	0.779		
Female	0552 (0.226–1.350)	0.193		
BMI, kg/m^2^	0.983 (0.925–1.045)	0.585		
Systolic BP, mmHg	0.988 (0.969–1.008)	0.227		
Diastolic BP, mmHg	0.972 (0.942–1.002)	0.063		
***Therapy***				
MRA	0.990 (0.521–1.884)	0.976		
ACE inhibitors	0.765 (0.341–1.720)	0.517		
ARB	1.231 (0.606–2.499)	0.566		
***Comorbidities***				
Diabetes mellitus	1.015 (0.471–2.188)	0.970		
Atrial fibrillation	0.861 (0.414–1.792)	0.689		
Coronary artery disease,	1.323 (0.587–2.979)	0.500		
***Laboratory parameters***				
NT-proBNP, pg/mL	2.449 (1.013–5.919)	0.047	2.247 (0.884–5.716)	0.089
GFR, ml/min/1.73 m^2^	1.000 (0.982–1.018)	0.996		
GGT, U/l	0.999 (0.994–1.003)	0.580		
eqAngiotensin I, pmol/L	1.831 (1.101–3.046)	0.020	1.134 (0.203–6.096)	0.888
eqAngiotensin II, pmol/L	2.244 (1.072–4.695)	0.032	0.919 (0.484–1.742)	0.795
Aldosterone, pmol/L	1.196 (0.568–2.519)	0.637		
AA2-Ratio, (pmol/L)/(pmol/L)	0.989 (0.962–1.016)	0.427		
PRA-S, pmol/L)	1.982 (1.122–3.503)	0.019	2.142 (1.203–3.816)	*p* = 0.010
ACE-S, (pmol/L)/(pmol/L)	0.858 (0.644–1.142)	0.272		
***Echo parameters***				
LV diameter, mm	1.005 (0.933–1.082)	0.902		
LVEF, %	0.964 (0.906–1.025)	0.240		
IVS, mm	0.960 (0.860–1.155)	0.960		
LV mass-index, g/m^2^	1.003 (0.988–1.018)	0.687		
LA length, mm	1.028 (0.973–1.087)	0.325		
RV diameter, mm	1.029 (0.978–1.083)	0.267		
RA length, mm	1.038 (0.986–1.093)	0.156		
E/A, ratio	0.958 (0.560–1.637)	0.874		
E/e’, ratio	1.081 (1.012–1.154)	0.020	1.081 (1.012–1.154)	0.020
TAPSE, mm	0.959 (0.865–1.063)	0.426		
sPAP, mmHg	1.012 (0.992–1.033)	0.227		

Continuous variables are shown as median and interquartile range, categorical variables are given as numbers and percentages. logarithmic values were used for the calculation of hazard ratios in all parameters of the renin angiotensin aldosterone system and NT-proBNP. HR indicates hazard ratio; CI, confidence interval; BMI, body mass index; BP, blood pressure; MRA, mineralocorticoid receptor antagonist; ACE, angiotensin- converting enzyme; ARB, angiotensin receptor blocker; NT-proBNP, N-terminal pro-brain natriuretic peptide; GFR, glomerular filtration rate; GGT, gamma-glutamyltransferase; AA2-Ratio, aldosterone-angiotensin II ratio; PRA-S, plasma renin activity; ACE-S, angiotensin converting enzyme activity; LV, left ventricle; LVEF, left ventricular ejection fraction; IVS, intraventricular septum thickness; LA, left atrium; RV, right ventricle; RA, right atrium and sPAP, systolic pulmonary artery pressure.

## Data Availability

Data including Appendix A is available and can be found online by clicking on the following link: https://figshare.com/s/b2a4c8711b7417722604, accessed on 14 December 2020.

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
