# Peer review of "Renin Feedback Is an Independent Predictor of Outcome in HFpEF"

_jpm, 2021, doi:10.3390/jpm11050370_

Round 1

Reviewer 1 Report

I had the pleasure to review the manuscript “Compensatory Renin Feedback in Response to RAAS Blockers in an Independent Predictor of Outcome in HFpEF” by Christina Binder et al. The authors performed an analysis of the impact of calculated circulating plasma renin (PRA-S) on mortality in a well-characterized registry of 150 patients with heart failure and preserved ejection fraction (HFpEF). They show that high PRA-S is independently associated with increased all-cause mortality, regardless of RAAS blocker therapy. This is a very interesting finding, as PRA-S seems to identify patients with higher long-term risk. The paper is well written and the English is clear. I have three main comments:

First, the rationale behind the title (“Compensatory Feedback”) and some parts of the discussion (“PRA-S remains high even under RAAS blocker therapy”) is not clear to me. The authors seem to assume that patients with high PRA-S at time of inclusion into the study may have had normal PRA-S values without RAAS inhibitors and they may have experienced a “compensatory feedback” due to the start of RAAS inhibitors. However, as far as I understood the article, PRA-S was assessed only once per patient. Therefore, it cannot be assumed that those patients with high PRA-S had normal PRA-S before RAAS blocker therapy. Those patients may have had higher baseline values of PRA-S than the remaining population with a similar reaction to RAAS inhibitor therapy. In my opinion, the title is also misleading as the study includes patients without RAAS blocker therapy. I would also suggest to rewrite the second sentence of the Discussion section in the abstract.

Second, please discuss the paper “Sympathetic and renin-angiotensin-aldosterone system activation in heart failure with preserved, mid-range and reduced ejection fraction” by Vergaro et al (International Journal of Cardiology 296 [2019] 91) as they seem to have analyzed similar parameters of the RAAS in a higher number of patients.

Third, please discuss and verify your method of PRA-S calculation even at the presence of RAAS inhibitor use.

Minor comments:

  • Please define the abbreviation “RAASi” in the abstract and throughout the text.
  • The page numbers are not continuous, please correct.
  • I would suggest to put the chapter 2.4 (echocardiography) between sections 2.1. and 2.2.
  • Please confirm that you actually used a stepwise-forward model for multivariate regression analysis. It seems to me that all parameters with p<0.05 in bivariate analysis were included into the multivariate analysis.
  • When during follow-up did patients develop exclusion criteria (aortic stenosis, reduced ejection fraction, amyloidosis)? If these diagnoses occurred within years after inclusion into this registry, I would suggest to include those patients, as such patients still represent a part of a typical HFpEF population presenting to a cardiology department.
  • I recommend not to use “Not surprisingly” and “Interestingly” in the Results section (lines 234 and 235; as well as “Not surprisingly” at line 253) and to leave the interpretion of the results to the discussion
  • Please remove “p=” in Table 4.
  • In my opinion, Figures 2-A, 2-B and 2-C are separate images, please consider numbering them 2, 3, and 4.

Reviewer 2 Report

Binder et al. present interesting findings that increased Renin activity despite RAAS therapy for some HFpEF patients may present with negative consequences. Indeed, high renin activity was linked to worsened survival for HFpEF patients and RASS inhibitors seemed to have limited effect in delaying mortality in HFpEF patients. Overall, the study design seemed well thought out and the authors did a good job in noting their study limitations (e.g. not monitoring for drug adherence). I only have minor comments that could likely be addressed in the discussion.

Comments:

  1. Figure 2 clearly demonstrates that there was variability in angI, ii PRA-S, etc. This would suggest genetic variability among these patients, yet the authors did not consider this in their discussion. Another possible reason that RAAS therapy had limited impact in this group is that some individuals (genetic variability) may be non-responders to the medication. It would have been interesting to know if this group had responders and non-responders. If so, would removal of non-responders show that RAAS therapy is effective in reducing mortality in the responder group? This would argue for the concept of precision medicine. Perhaps, knowing a patients PRA-S profile would indicate whether RAAS therapy is warranted for the HFpEF population if mortality is the only important variable.
  2. While mortality was assessed in this patient population, it's unclear if RAAS therapy improved quality of life or if again increased PRA-S led to worsened quality of life outcomes. While the goal is to improve mortality, quality of life is an equally important issue. RAAS therapy may not help with survival in HFpEF patients but it may still lead to overall daily living.  

Round 2

Reviewer 1 Report

The manuscript has substantially improved.